# Prevalence and Genetic Diversity of Toxin Genes in Clinical Isolates of *Clostridium perfringens*: Coexistence of Alpha-Toxin Variant and Binary Enterotoxin Genes (*bec*/*cpile*)

**DOI:** 10.3390/toxins11060326

**Published:** 2019-06-06

**Authors:** Asami Matsuda, Meiji Soe Aung, Noriko Urushibara, Mitsuyo Kawaguchiya, Ayako Sumi, Mayumi Nakamura, Yuka Horino, Masahiko Ito, Satoshi Habadera, Nobumichi Kobayashi

**Affiliations:** 1Department of Hygiene, Sapporo Medical University School of Medicine, Sapporo 060-8556, Japan; a-matsuda@sapmed.ac.jp (A.M.); noriko-u@sapmed.ac.jp (N.U.); kawaguchiya@sapmed.ac.jp (M.K.); sumi@sapmed.ac.jp (A.S.); nkobayas@sapmed.ac.jp (N.K.); 2Sapporo Clinical Laboratory, Inc., Sapporo 060-0005, Japan; saturin-saikin@saturin.co.jp (M.N.); yu_nonentity@yahoo.co.jp (Y.H.); m-ito@saturin.co.jp (M.I.); s-habadera@saturin.co.jp (S.H.)

**Keywords:** *Clostridium perfringens*, alpha-toxin, enterotoxin, beta2 toxin, NetB, BEC/CPILE, toxinotype, ST (sequence type)

## Abstract

*Clostridium perfringens* (*C. perfringens*) is responsible for food-borne gastroenteritis and other infectious diseases, and toxins produced by this bacterium play a key role in pathogenesis. Although various toxins have been described for *C. perfringens* isolates from humans and animals, prevalence of individual toxins among clinical isolates has not yet been well explored. In the present study, a total of 798 *C. perfringens* clinical isolates were investigated for prevalence of eight toxin genes and their genetic diversity by PCR, nucleotide sequencing, and phylogenetic analysis. Besides the alpha-toxin gene (*plc*) present in all the isolates, the most common toxin gene was *cpe* (enterotoxin) (34.2%), followed by *cpb2* (beta2 toxin) (1.4%), *netB* (NetB) (0.3%), and *bec*/*cpile* (binary enterotoxin BEC/CPILE) (0.1%), while beta-, epsilon-, and iota-toxin genes were not detected. Genetic analysis of toxin genes indicated a high level of conservation of *plc*, *cpe*, and *netB*. In contrast, *cpb2* was revealed to be considerably divergent, containing at least two lineages. Alpha-toxin among 46 isolates was classified into ten sequence types, among which common types were distinct from those reported for avian isolates. A single isolate with *bec*/*cpile* harbored a *plc* variant containing an insertion of 834-bp sequence, suggesting its putative origin from chickens.

## 1. Introduction

*Clostridium perfringens* (*C. perfringens*) is a Gram-positive, spore-forming anaerobic bacterium that causes mainly toxic infectious diseases in humans and animals represented by myonecrosis (gas gangrene) and gastroenteritis [1]. The pathogenicity of this bacterium is associated with extracellular toxins, which include four major toxins (alpha-, beta-, epsilon-, iota-toxin) and enterotoxin. All the *C. perfringens* isolates produce alpha-toxin, which is implicated in myonecrosis in humans and necrotic enteritis in chickens. Beta-, epsilon-, and iota-toxins are responsible for gastrointestinal diseases or enterotoxemia mostly in non-human animals. *C. perfringens* enterotoxin (CPE) is an essential factor of human food-poisoning, as well as antibiotic-associated and nosocomial diarrheal diseases [2]. Recently, a novel enterotoxin designated as BEC (binary enterotoxin of *C. perfringens*) or CPILE (*C. perfringens* iota-like enterotoxin) were reported in non-CPE producing strains derived from food poisoning outbreaks in Japan [3,4]. Moreover, NetB and beta2 toxins have been described as those associated with necrotic enteritis in chickens and enteric diseases in humans and animals, respectively [5,6].

*C. perfringens* strains have been conventionally classified into five toxinotypes (A–E) based on production of four major toxins, which is important for epidemiology and diagnosis of diseases caused by this bacterium because different diseases are caused by different toxinotypes [1]. However, this scheme does not include CPE and other newly described toxins for classification. Accordingly, Rood and coworkers proposed a modified scheme of toxinotype which incorporated CPE and NetB for classification criteria [7]. In this scheme, strains that produce alpha-toxin and CPE, and alpha-toxin and NetB, without beta-, epsilon-, and iota-toxins, are assigned to type F and G, respectively. Nevertheless, a toxinotype scheme taking into account BEC/CPILE and beta2 toxin has not yet been designed.

Prevalence of *C. perfringens* toxins in human diseases has been described mostly for outbreaks and sporadic cases of intestinal diseases. *C. perfringens* type A strains producing CPE (type F) are responsible for 5%–20% of all antibiotic-associated diarrhea and non-food-borne gastrointestinal diseases [1,8]. Among *C. perfringens* isolates from human diarrheal diseases, detection rates of CPE gene (*cpe*) were described as 4.9%–7.8% in different countries [9,10,11]. However, prevalence of *C. perfringens* toxins (particularly other than CPE) in human clinical isolates has not yet been systematically studied, although a low prevalence (0.78%) of BEC/CPILE (*bec/cpile*) was shown in only one report in Japan [11]. Beta2 toxin was reported for its potential association with autistic children [12,13], while its etiologic significance is still obscure because of its prevalence among healthy individuals [14]. Besides, genetic diversity has been analyzed for alpha-toxin, revealing presence of at least ten alpha-toxin types and subtypes in avian isolates [15,16] and also a variant containing a long insertion sequence [17]. However, diversity of alpha-toxin as well as other toxins in human *C. perfringens* isolates remains undetermined.

In the present study, prevalence of eight toxin genes of *C. perfringens* was analyzed for a number of isolates from clinical specimens collected in medical facilities in Hokkaido, the northern main island of Japan. Detected toxin genes were analyzed phylogenetically, along with genotyping of isolates, to elucidate evolutionary status of toxin genes as well as clonality of isolates having specific toxins. Consequently, substantial genetic diversity was revealed for beta2 toxin, while a highly conserved nature was observed for alpha-toxin and CPE. Remarkably, an isolate with *bec/cpile*, which was derived from a sporadic gastroenteritis case, harbored a distinctive genetic variant of *plc* that had been reported only in chickens, suggesting its relevance to avian *C. perfringens*.

## 2. Results

### 2.1. Prevalence of Toxin Genes and Toxinotypes

Toxin genes in *C. perfringens* isolates from individual clinical specimens and the patient groups are summarized in Table 1. While fecal isolates accounted for 97%, some isolates were derived from specimens not related to intestines (i.e., pus, urine, venous blood). Except for alpha-toxin gene (*plc*) detected in all the isolates, the highest prevalence was found for *cpe* (34.2%), followed by *cpb2* (1.4%). *netB* and *bec/cpile* were identified in two (0.3%) and one (0.1%) isolate from feces, respectively. beta-, epsilon-, iota-toxin were not detected. *cpe* was detected in 33.7% of fecal isolates (262/777) and all the isolates from pus, urine, blood, and intestinal mucosa. The *cpe*-positive rate was significantly higher in male patients than female patients, and also in outpatients than inpatients (*p* < 0.01).

Most of isolates were assigned to toxinotype A (64.8%) and F (33.5%) (Table 2). Two *netB*-positive isolates were classified into type G, according to the scheme proposed by Rood et al. [7]. In the present study, we created provisional toxinotypes H1, H2, and I, which represent *plc*- and *cpb2*-positive, *plc*-, *cpe*-, and *cpb2*-positive, and *plc*- and *bec/cpile*-positive isolates, respectively. Five, six, and one isolate were assigned to types H1, H2, and I, respectively.

### 2.2. Sequence Type (ST) Based on MLST (multilocus sequence typing) Scheme 

A total of 46 *C. perfringens* isolates were analyzed genetically (Table 3). They included all the *cpb2*-, *netB*-, and *bec/cpile*-positive isolates (n = 14), and the remaining 32 isolates (31 *cpe*-positive isolates) were selected from different clinical specimens and patients.

The sequence type (ST) of the 46 isolates was determined based on the scheme of allele and profile definitions described by Xiao et al. [18]. Among the 46 isolates analyzed, ST was determined for 26 isolates, while others were untypeable and were described as new types that included those related to known STs, such as SLV, DLV, and TLV (single-, double-, and triple-locus variant, respectively) (Table 3). ST41 and its SLV or DLV were commonly found for type A and type F (*cpe*-positive) *C. perfringens* isolates. Eleven *cpb2*-positive isolates were classified into seven STs, among which ST36 was assigned to four type H2 (*cpe*- and *cpb2*-positive) isolates. Two *netB*-positive isolates (type G) belonged to ST21, and a *bec/cpile*-positive isolate CP653 was assigned to a new ST. An isolate from blood (CP322) was classified into a unique type ST8.

### 2.3. Genetic Analysis of Toxin Genes

*C. perfringens* toxin genes detected in this study, i.e., *plc* (n = 46), *cpe* (n = 30), *cpb2* (n = 11), *netB* (n = 2), and *bec/cpile* (n = 1), among the selected 46 isolates were analyzed for their lineages and sequence diversity together with toxin gene sequences available in the GenBank database.

#### 2.3.1. Alpha-Toxin (Phospholipase C) Gene (plc) and Alpha-Toxin Type

*plc* of 46 isolates showed extremely high sequence identity (>97.8%) to each other and that of human strain 13 [19]. For all the isolates, alpha-toxin type was determined according to the scheme described by Sheedy et al. [15] and Abildgaard et al. [16]. Based on the definition of amino acids at positions 13, 54, 202, 205, and 373, all the isolates were first grouped into type I, IV, V, and VIII. Among these toxin types, types I, V, and VIII that had been described in the previous studies for avian isolates [15,16] were assigned to six isolates, while the remaining isolates had different amino acids at seven other positions, indicating that they do not belong to previously reported types. Therefore, for these alpha-toxin sequences, the new subtypes IVb, IVc, IVd, Vd, Ve, Vf, and Vg were assigned (Appendix A). The most prevalent alpha-toxin type was Ve, followed by IVb, Vd, and IVc, which were all new types and accounted for 80% (37/46). In the alpha-toxin amino acid (aa) sequence, the 11 positions of aa diversity are located outside the N-terminal catalytic region (central loop domain) (Appendix A) [20].

*plc* of CP653, which was the only isolate carrying *bec/cpile*, was proved to be a variant gene having an additional 834-bp sequence inserted into *plc*, after nucleotide no. 340 (Appendix A). Because of this insertion, CP653 *plc* was presumed to have two ORFs (open reading frames) in N- and C-terminal portions (Appendix A, putative ORF(1)). By using BLAST (Basic Local Alignment Search Tool) sequence homology search, a similar *plc* variant with internal insertion was detected in strain S03 from healthy chicken (GenBank accession no. EU839836) [16], showing 99.85% identity to CP653 *plc*, having the same genetic structure as that of avian *C. perfringens* strain CPBC16ML [17]. The 834-bp sequence was identical to that reported as group II intron (GenBank accession no. DQ787115) (Appendix A) [16,17]. In the study of strain CPBC16ML, the inserted sequence was revealed to be spliced during transcription, and intact alpha-toxin product was produced from this variant [17]. Accordingly, supposing that CP653 *plc* produces intact alpha-toxin by splicing similarly to strains CPBC16ML and S03 (Appendix A, putative ORF(2)), deduced alpha-toxin aa sequences of CP653 and S03 were highly similar with only two aa differences (Appendix A). Moreover, the inserted 834-bp sequence of CP653 (group II intron) exhibited high sequence identity (94%–96%) to a portion of *C. perfringens* plasmids by BLAST search and contained no ORF (Appendix A). On both sides of the inserted portion of intact *plc* (5’- and 3’-sides of nucleotide 340) and in the inserted sequence-like region in the plasmids, similar sequences with 10- and 6-nucleotides, respectively, were detected (GAT…AGTTGGT and GCTATT in the 5’- and 3’-side, respectively) (Appendix A).

#### 2.3.2. Enterotoxin Gene (cpe)

Among the 31 *cpe*-positive isolates, most *cpe* genes (28 isolates) were located on plasmid, which was mapped in the IS*1151* locus (20 isolates) or IS*1470*-like locus (8 isolates) (Table 3). *cpe* was highly conserved showing >95.6% nucleotide sequence identity. Phylogenetic analysis revealed the presence of two distinct lineages designated major and minor lineages (Appendix A). All the *cpe* from isolates in our present study were included in the major lineage, among which sequence identity was 99.9%–100%. CPE of only one isolate, CP859, had a single aa (nucleotide) difference from all other isolates in domain III (Appendix A). Although divergent aa sites in CPE of major and minor lineages were scattered over the protein, aa residues 44–116 associated with large complex formation (transmembrane domain forming pores) [21,22], aa involved in binding to claudin-3/-4 [23], and the C-terminal receptor binding region [24] were mostly conserved among strains.

#### 2.3.3. Beta2 Toxin Gene (cbp2)

Two lineages (cluster 1 and 2) were identified in phylogenetic tree of beta2 toxin gene (*cpb2*) (Figure 1) and distinguishable from prototype of beta2 toxin in strain CWC245 [25]. *cpb2* of nine isolates in the present study was included in cluster 1, while one isolate (CP458) was assigned to cluster 2, and the remaining isolate (CP412) was closely related to strain CWC245. The nucleotide sequence identity between clusters 1 and 2 was 71.9%–74.8%, in contrast to >95% within each cluster (Appendix A).

*cpb2* of both clusters 1 and 2 showed relatively low identity to that of strain CWC245 and CP412 (91.7%–94.9% and 71.0%–74.8%, respectively). Considerable aa sequence diversity of beta2 toxin was observed, especially in the N-terminal, approximately at the 100 aa-sequence (Appendix A).

#### 2.3.4. NetB Toxin Gene (netB)

*netB* was detected in two strains, CP201 and CP238, both of which had type I alpha-toxin and belonged to ST21. These *netB* sequences were identical and showed >99% identity to *netB* sequences available in GenBank database, including the *netB* prototype in strain EHE-NE18 (accession no. EU143239).

#### 2.3.5. Binary Enterotoxin Gene (becAB/cpile-ab)

*becA/cpile-a* and *becB/cpile-b* of *C. perfringens* isolate CP653 showed 98.5% and 97.9% nucleotide sequence identity, respectively (98.3% and 97.1% aa identity, respectively) to the reported sequences of the binary enterotoxin (strains TS1, OS1, and W5052) [3,4] which are absolutely identical. BECa/CPILE-a sequence of CP653 was different from those of three reported strains by seven aa, of which five aa were located in the N-terminal region (Appendix A). Among CP653 and the three reported strains, common structural traits (motifs) in ADP-ribosylating toxin-family proteins [26], regions associated with NADH recognition, and binding to actin [27] were conserved.

## 3. Discussion

In the present study, prevalence of various toxin genes in *C. perfringens* was revealed for isolates from clinical specimens, most of which were feces derived from sporadic diarrheal cases. CPE has been implicated in gastroenteritis in humans and documented to be produced by 2%–8% of *C. perfringens* isolates from diarrhea in the community [8,10,11,28]. Similar prevalence of *cpe* (4%) was reported for isolates from retail meat [29]. In contrast, CPE has been shown as an essential virulence factor in food-borne diarrheal outbreaks due to *C. perfringens* [30,31], and similarly high detection rates of *cpe* (40%–70%) were described for gastroenteritis outbreaks that occurred mostly in facilities [32,33]. In our study, the detection rate of 34.2% (33.7% in isolates from feces) from sporadic cases was higher than those reported for similar study settings previously [11,28]. A study on *C. perfringens* from food poisoning outbreaks in Japan indicated a higher frequency of plasmid-mediated *cpe* and predominance of *cpe* with a downstream IS*1151* sequence [30]. Similarly, in our study, most of the *cpe*-positive isolates analyzed had *cpe* on plasmid, among which IS*1151*-associated *cpe* was dominant (71%, 20/28). Accordingly, although isolates from evident food-borne outbreaks were not included in our study, it is conceivable that a portion of isolates might be derived from small-scale outbreaks within families or facilities. It was of note that *cpe* was detected in all isolates from pus, urine, and venous blood (n = 6), some of which carried plasmid with *cpe*-IS*1151* locus. Despite the still low number of isolates from non-fecal/intestinal specimens, *cpe* on plasmid is suggested to be involved in extraintestinal disease due to *C. perfringens*.

BEC/CPILE is a novel enterotoxin consisting of two proteins, among which the smaller component (BECa/CPILE-a) has ADP-ribosylating activity on actin [3,4]. This toxin was identified in *C. perfringens* without having *cpe* which caused food poisoning outbreaks in Japan [34]. Although little is known about the prevalence of *bec/cpile* in *C. perfringens* isolates, a single survey indicated 0.78% positivity of *bec/cpile* among isolates from diarrheal patients, which was much lower than that of *cpe* (7.8%) [11]. Similarly, considerably low prevalence of *bec/cpile* (one isolate, 0.1%) in contrast to *cpe* (34.2%) was observed in our present study. While previously reported sequences of *becAB/cpile-ab* in three *C. perfringens* strains were absolutely identical [3,4], these genes detected in isolate CP653 in our study showed slight diversity, suggesting occurrence of genetic evolution via multiple replication and spread in nature. Because the divergent aa sites in the BEC-a/CPILE-a sequence of CP653 were located outside the functionally important sites and regions (Appendix A), this variant protein is suggested to possess enterotoxic activity.

Remarkably, isolate CP653 was revealed to have a *plc*-variant containing internal 834-bp insertion, which is virtually identical to that has been reported in only four strains (CPBC16ML, S03, and other two strains) from healthy chickens to date [16,17], suggesting strong association with chicken. In contrast, CP653 is the first *C. perfringens* with the *plc*-variant isolated from a human. Detection of the *plc*-variant in *bec/cpile*-positive human isolate CP653 may indicate possibility that *bec/cpile* might have been of avian origin. To verify it, it is necessary to investigate the presence of *bec/cpile* among avian isolates, particularly *C. perfringens* with the *plc* variant. While the mechanism by which the 834-bp sequence was inserted into *plc* has not yet been elucidated, homologous short sequences were identified in both sides of insertion site of *plc* as well as specific sites on *C. perfringens* plasmid enclosing the 834-bp insertion sequence. This finding suggests that the *plc*-variant might be generated via template switching of DNA polymerase during nucleotide synthesis, which was also presumed as a mechanism of internal deletion of bacterial genes in *Staphylococcus aureus* [35,36].

Alpha-toxin types assigned to 46 isolates analyzed in the present study were mostly new types, which had not been reported for avian isolates [15,16]. Thus, *C. perfringens* clones distributed to humans and animals may be considered essentially different. Genetic difference of *C. perfringens* between human and animal isolates has been documented by other studies [31,37]. However, *netB*, which is associated with avian necrotic enteritis [5], was detected in *C. perfringens* isolates from healthy human subjects [37], and isolates from feces with gastroenteritis patients in the present study. These findings suggest the occurrence of interspecies transmission of *C. perfringens* and a role of humans as a source of pathogenic clones to chickens.

Genetic analysis indicated that *plc* and *cpe* in isolates of our present study and those reported previously showed extremely high sequence identity, and functionally critical motifs, structures, and amino acids for toxic activity in alpha-toxin and CPE were absolutely conserved. In contrast, genetic diversity of *cpb2* was remarkable, showing three distinct clusters in the present study. Similarly, diversity of beta2 toxin was described previously [31,38]. Beta2 toxin was first identified in *C. perfringens* isolated from a piglet with necrotizing enteritis [25], showing high detection rate in diarrheic pigs [6,39], and also ascribable to intestinal diseases in horses, dogs, and calves [25,40]. However, wide spread of beta2 toxin gene was documented in healthy calves and pigs [41,42]. In humans, *cpb2* was detected in food poisoning cases [31] as well as healthy individuals [14], and also in children with autism at a higher rate than control children [12]. In the present study, despite at a low rate, *cpb2* was detected in 11 isolates from feces, with six isolates being associated with *cpe*. Because *cpb2* was detected in isolates with seven different STs, *cpb2*-positive isolates were considered to be multiclonal, while *cpb2* of cluster 1 was the most common. Although pathogenic significance beta2 toxin in humans is not evident, it is presumable that predisposing factors in host are associated with toxic activity of beta2 toxin as reviewed by Schotte et al. [42]. Since considerable genetic diversity in this toxin is evident, genetic grouping of *cpb2*, i.e., two clusters and prototype-related group, classified in the present study may be introduced into epidemiological study to relate this gene in prevalence in animals and humans, and their health status or endogenous factors. Further study is necessary to elucidate etiological significance of *cpb2* carried by *C. perfringens* in humans.

## 4. Materials and Methods

### 4.1. C. perfringens Isolates

A total of 798 non-duplicate *C. perfringens* isolates recovered from clinical specimens during a period between January 2017 and March 2018 were analyzed. Clinical specimens were collected in hospitals and clinics in Hokkaido prefecture, Japan, and sent to the Sapporo Clinical Laboratory, Inc. for isolation and identification of bacterial species. For bacterial isolation, specimens were cultured in GAM Semi-Solid medium (Nissui) anaerobically at 35 °C for 24 h, followed by sequential anaerobic culture on 5% sheep blood agar (Nippon Becton Dickinson). Colonies on agar plates were analyzed for the species by MALDI-TOF Mass Spectrometry using MALDI Biotyper (BRUKER). All isolates were stored in Microbank (Pro-lab Diagnostics, Richmond Hill, Canada) at −80 °C. For analysis of isolates, the stored isolates were recovered in GAM broth (Nissui) at 35 °C for 24 h.

### 4.2. Genetic Analysis of C. Perfringens

*C. perfringens* toxin genes encoding alpha-, beta-, epsilon-, and iota-toxin, and enterotoxin were detected by multiplex PCR as described previously [43]. For further PCR detection of beta2 toxin gene (*cpb2*), NetB gene (*netB*), and BEC/CPILE gene (*bec/cpile*), previously reported primers were used [7,11,44]. For the selected isolates with *cpe*, multiplex PCR assay was used to specify location of *cpe* in chromosome or plasmid (locus with IS*1470*-like or IS*1151* sequence) [45].

Nucleotide sequences of toxin genes were determined by direct sequencing with PCR products, using the BigDye Terminator v3.1 Cycle Sequencing Kit (Applied Biosystems, Foster City, CA) on an automated DNA sequencer (ABI PRISM 3100). Primers designed in the present study to amplify individual toxin genes are listed in Appendix A. Phylogenetic dendrograms of toxin genes *cpe* and *cpb2* were constructed by maximum likelihood method using the MEGA7 software package. Full length sequences (ORF) of these genes determined in the present study were used for phylogenetic analysis. Reference sequences of *cpe* and *cpb2* to be included in phylogenetic trees were obtained from GenBank database, although only partial *cpb2* sequences were available for 16 strains. Multiple alignment of nucleotide/amino acid sequences determined in the present study and those retrieved from the GenBank database was performed by Clustal Omega program (https://www.ebi.ac.uk/Tools/msa/clustalo/). Sequence identity was calculated by using LALIGN sever (https://embnet.vital-it.ch/software/LALIGN_form.html).

For selected isolates, ST was determined according to the Xiao’s scheme of MLST [18] employing eight loci (*gyrB*, *sigK*, *sod*, *groEL*, *pgk*, *nadA*, *colA*, and *plc*). Sequence data were submitted to a web-based tool (https://pubmlst.org/bigsdb?db=pubmlst_cperfringens_seqdef_xiao) to obtain genotypes of individual loci and ST. Although a web tool of new MLST locus/sequence definition for *C. perfringens* (https://pubmlst.org/bigsdb?db=pubmlst_cperfringens_seqdef) has been created under PubMLST website, the database in this scheme has not yet been completed. Thus, we employed Xiao’s MLST scheme which is available for legacy purposes to understand the clonal diversity of isolates.

### 4.3. Classification of Toxinotype and Alpha-Toxin Type

Toxinotypes A to E of *C. perfringens* have been conventionally classified based on production of major four toxins (alpha-, beta-, epsilon-, iota-toxin); type A strain produces only alpha-toxin, while type B, C, D, E strains produce alpha-, beta- and epsilon-toxins, alpha- and beta-toxins, alpha- and epsilon-toxins, and alpha- and iota-toxins, respectively [46]. In 2018, Rood and coworkers [7] proposed a modified typing scheme and defined two new toxinotypes F and G which represent strains producing alpha-toxin and enterotoxin, and alpha- and NetB-toxins, respectively. In the present study, toxinotypes were classified based on the modified scheme [7]. Accordingly, type A did not include isolates producing CPE, although traditionally, strain with alpha-toxin and CPE had been referred to as “type A strain producing CPE”. Furthermore, some provisional toxinotypes were created to show toxin profiles of isolates which could not be assigned to toxinotypes A to G.

Alpha-toxin type was classified by the definitions described by Sheedy et al. [15] and Abildgaard et al. [16] based on different amino acids compared with those of strain 13. Amino acid substitutions of four types detected in the present study are as follows: type I, T13A; type IV, T13A, L54M; type V, T13A, D202A, A205T; type VIII, I373V. Subtypes of IV (IVb, IVc, IVd) and V (Vd, Ve, Vf, Vg) which have not been described previously were created in the present study based on amino acid difference at positions 57, 166, 244, 362, 363, 365, and 373. Subtype names were assigned arbitrarily.

### 4.4. GenBank Accession Numbers

The nucleotide sequences of *plc*, *cpe*, *cpb2*, *netB*, *bec/cpile* were deposited in the GenBank database under accession numbers listed in Appendix A.

## Figures and Tables

**Figure 1 toxins-11-00326-f001:**
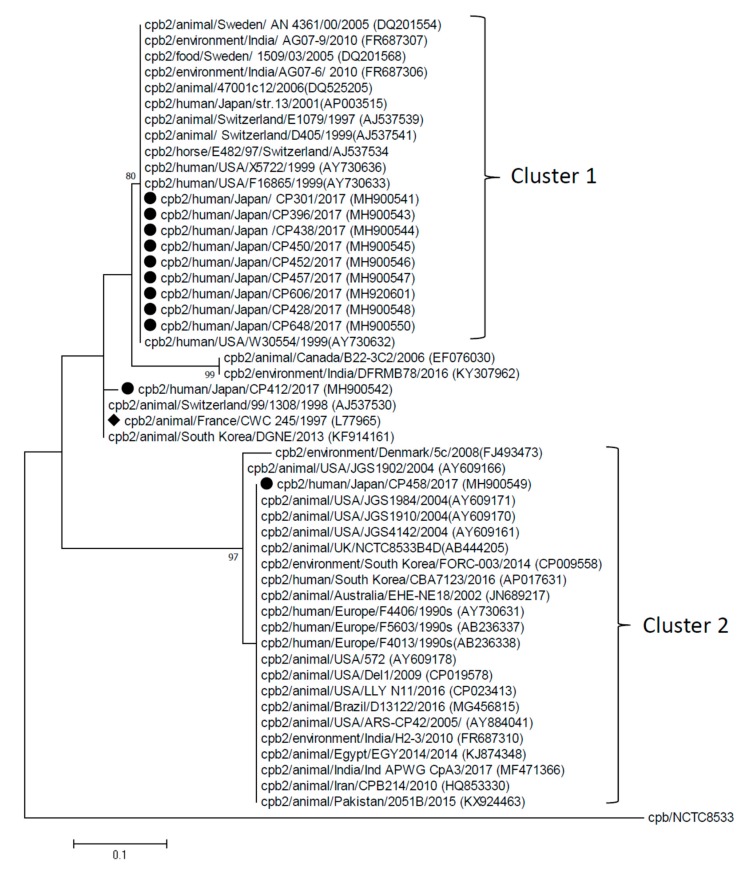
Phylogenetic dendrogram of *cpb2* constructed by maximum likelihood method using MEGA7 version 7.0.26. The tree was statistically supported by bootstrapping with 1000 replicates, and genetic distances were calculated by the Kimura two-parameter model. The beta-toxin gene (*cpb*) of strain NCTC8553 (GenBank accession no. KP064408) was included as outgroup. A variation scale is provided at bottom. The percentage bootstrap support is indicated by values at each node (values <80 are omitted). Isolates analyzed in the present study are shown with closed circles, while a diamond indicates strain CWC245 which has the prototype of *cpb2*. Clusters are shown on the right.

**Table 1 toxins-11-00326-t001:** Prevalence of *plc*, *cpe*, *cpb2*, *netB*, and *bec/cpile* gene among *C. perfringens* isolates from different types of specimens and patients.

Specimens/Patients /Infection Types	Total no. of Isolates	Number of Isolates with Toxin Gene (%)				
		*plc*	*cpe*	*cpb2*	*netB*	*bec/cpile*
Specimens						
Feces	777	777	262	11	2	1
Intestinal fluid	8	8	2	0	0	0
Bile/gallbladder	5	5	1	0	0	0
Pus	4	4	4	0	0	0
Urine	1	1	1	0	0	0
Venous blood	1	1	1	0	0	0
Intestinal mucosa	2	2	2	0	0	0
Total	798	798 (100%)	273 (34.2%)	11 (1.4%)	2 (0.3%)	1 (0.1%)
Patients						
male	370	370 (100%)	155 (41.9%) ^1^	5 (1.4%)	1 (0.3%)	0 (0%)
female	428	428 (100%)	118 (27.6%)	6 (1.4%)	1 (0.2%)	1 (0.2%)
Inpatient	499	499 (100%)	90 (18%)	7 (1.4%)	1 (0.2%)	0 (0%)
Outpatient	299	299 (100%)	183 (61.2%) ^1^	4 (1.3%)	1 (0.3%)	1 (0.3%)

^1^*p* < 0.01.

**Table 2 toxins-11-00326-t002:** Toxinotypes of *C. perfringens* isolates (n = 798).

Toxinotype ^1^	Alpha-Toxin (*plc*)	Beta-Toxin (*cpb*)	Epsilon-Toxin (*etx*)	Iota-Toxin (*iA*)	Enterotoxin (*cpe*)	NetB (*netB*)	Beta2-Toxin (*cpb2*)	BEC/CPILE (*bec*/*cpile*)	No. of Isolates (%)
A	+	−	−	−	−	−	−	−	517 (64.8%)
B	+	+	+	−	−	−	−	−	0
C	+	+	−	−	± ^2^	−	−	−	0
D	+	−	+	−	±	−	−	−	0
E	+	−	−	+	±	−	−	−	0
F	+	−	−	−	+	−	−	−	267 (33.5%)
G	+	−	−	−	−	+	−	−	2 (0.3%)
H1	+	−	−	−	−	−	+	−	5 (0.6%)
H2	+	−	−	−	+	−	+	−	6 (0.8%)
I	+	−	−	−	−	−	−	+	1 (0.1%)

^1^ New toxinotypes F and G were defined previously [7]. H1, H2, and I were provisionally created in the present study, hence unapproved toxinotypes. ^2^ ± represents either positive or negative for cpe.

**Table 3 toxins-11-00326-t003:** Sequence type (ST) and toxin gene profiles of 46 *C. perfringens* isolates.

Strain ID	Sex	Age	Inpatient /Outpatient	Specimen	Alpha-Toxin Sequence Type	Toxin Gene (+ *plc*)	Location of *cpe*	ST (Closest ST for New Alleric Profile)	Allelic Profile
CP201	M	80	Outpatient	feces	I	*netB*	-	ST21	3-1-3-4-4-3-2-1-1
CP238	F	83	Inpatient	feces	I	*netB*	-	ST21	3-1-3-4-4-3-2-1-1
CP606	F	83	Inpatient	feces	I	*cpb2* (C-1) ^1^	-	New: ST21 (TLV)	10-1-1-9*-3-2-1-1
CP501	M	1	Outpatient	feces	IVb	*cpe*	Plasmid (IS1151-locus)	New	4-30-19-3-1-3-3-1
CP322	F	86	Outpatient	venous blood	IVb	*cpe*	Plasmid (IS1151-locus)	ST8	4-5-1-3-5-4-3-3
CP282	M	1	Outpatient	feces	IVb	*cpe*	Plasmid (IS1151-locus)	New	4-30-19-3-1-3-3-1
CP904	M	41	Outpatient	feces	IVb	*cpe*	Plasmid (IS1151-locus)	ST25	17-3-1-3-1-3-3-14
CP648	F	89	Inpatient	feces	IVb	*cpb2* (C-1)	-	New: ST25 (TLV)	4-9-5-3-1-3-3-3
CP628	F	89	Inpatient	feces	IVb	-	-	New: ST25 (TLV)	4-3-5-3-1-3-3-3
CP281	F	5	Outpatient	bile	IVb	*cpe*	Plasmid (IS1151-locus)	ST25	17-3-1-3-1-3-3-14
CP458	F	62	Outpatient	feces	IVb	*cpe, cpb2* (C-2)	Plasmid (IS1151-locus)	New	4-30-19-3-1-3-3-1
CP457	M	82	Inpatient	feces	IVb	*cpe, cpb2* (C-1)	Plasmid (IS1470-like locus)	ST36	6-6-1-7-1-5-4-1
CP428	F	78	Inpatient	feces	IVc	*cpe, cpb2* (C-1)	Plasmid (IS1470-like locus)	ST36	6-6-1-7-1-5-4-1
CP438	F	78	Inpatient	feces	IVc	*cpe, cpb2* (C-1)	Plasmid (IS1470-like locus)	ST36	6-6-1-7-1-5-4-1
CP450	F	68	Inpatient	feces	IVc	*cpe, cpb2* (C-1)	Plasmid (IS1470-like locus)	ST36	6-6-1-7-1-5-4-1
CP157	M	72	Inpatient	pus	IVd	*cpe*	NT	New	4-30-19-3-1-3-3-1
CP714	M	81	Outpatient	pus	V	*cpe*	Plasmid (IS1151-locus)	ST41	19-5-1-5-5-2-3-1
CP885	F	93	Inpatient	feces	V	*cpe*	Plasmid (IS1151-locus)	ST41	19-5-1-5-5-2-3-1
CP616	F	91	Inpatient	large intestine mucous membrane	Vd	*cpe*	Plasmid (IS1151-locus)	ST7	5-4-1-5-4-3-3-3
CP672	F	46	Outpatient	feces	Vd	*cpe*	Plasmid (IS1151-locus)	ST41	19-5-1-5-5-2-3-1
CP792	F	51	Outpatient	feces	Vd	*cpe*	Plasmid (IS1151-locus)	New: ST41 (SLV)	19-5-1-5-1-2-3-1
CP667	M	86	Inpatient	feces	Vd	*cpe*	Plasmid (IS1151-locus)	ST41	19-5-1-5-5-2-3-1
CP489	M	1	Outpatient	feces	Ve	-	-	New: ST41 (SLV)	19-5-1-5-5-24-3-1
CP88	F	0	Outpatient	feces	Ve	-	-	New: ST41 (DLV)	19-10-1-5-9-2-3-1
CP412	M	68	Outpatient	feces	Ve	*cpb2*	*-*	New: ST41 (TLV)	4-5-3-5-5-2-6-1
CP757	F	101	Inpatient	feces	Ve	*cpe*	Plasmid (IS1151-locus)	ST41	19-5-1-5-5-2-3-1
CP734	F	20	Outpatient	feces	Ve	-	-	ST41	19-5-1-5-5-2-3-1
CP666	F	96	Inpatient	feces	Ve	*cpe*	Plasmid (IS1151-locus)	New: ST41 (SLV)	19-5-1-5-1-2-3-1
CP833	M	77	Outpatient	pus	Ve	*cpe*	Plasmid (IS1151-locus)	ST41	19-5-1-5-5-2-3-1
CP832	M	4	Outpatient	feces	Ve	*cpe*	Plasmid (IS1151-locus)	ST41	19-5-1-5-5-2-3-1
CP782	M	25	Outpatient	feces	Ve	*cpe*	Plasmid (IS1470-like locus)	ST41	19-5-1-5-5-2-3-?1
CP706	M	16	Outpatient	feces	Ve	-	-	New	19-3-19-5-4-5-3-1
CP91	F	82	Outpatient	urine	Ve	*cpe*	NT	New: ST41 (SLV)	3-5-1-5-5-2-3-1
CP396	M	2	Outpatient	feces	Ve	*cpe, cpb2* (C-1)	Plasmid (IS1470-like locus)	ST28	19-19-13-5-5-2-3-1
CP859	F	95	Outpatient	feces	Ve	*cpe*	Plasmid (IS1151-locus)	New: ST41 (SLV)	1-5-1-5-5-2-3-1
CP301	M	70	Inpatient	feces	Ve	*cpb2* (C-1)	*-*	ST28	19-19-13-5-5-2-3-1
CP232	F	89	Inpatient	bile	Ve	*cpe*	Plasmid (IS1151-locus)	ST41	19-5-1-5-5-2-3-1
CP94	M	69	Outpatient	intestinal fluid	Ve	-	-	New: ST41 (SLV)	19-5-3-5-5-2-3-1
CP570	M	3	Outpatient	feces	Ve	*cpe*	Plasmid (IS1470-like locus)	ST41	19-5-1-5-5-2-3-1
CP502	F	2	Outpatient	feces	Ve	*cpe*	Plasmid (IS1151-locus)	ST41	19-5-1-5-5-2-3-1
CP228	F	58	Outpatient	intestinal fluid	Ve	*cpe*	NT	ST41	19-5-1-5-5-2-3-1
CP327	F	50	Outpatient	intential wall	Ve	*cpe*	Plasmid (IS1151-locus)	ST41	19-5-1-5-5-2-3-1
CP422	F	90	Outpatient	feces	Ve	*cpe*	Plasmid (IS1470-like locus)	New: ST41 (SLV)	1-5-1-5*-5-2-3-1
CP400	M	NA	Outpatient	feces	Vf	-	-	New	4-8-1-1-8-7-4-1
CP653	F	44	Outpatient	feces	Vg	*bec* (*cpile*) *plc*-variant ^2^	-	New	4-1-19*-1-3-5-1-1
CP452	M	26	Outpatient	feces	VIII	*cpb2* (C-1)	-	New	4-30-1-9*-3-12-8-11

NT, nontypeable; NA, not available; SLV (single locus variant), TLV (triple locus variant). * Sequences of these allelic loci have a few nucleotide diversity (closest locus numbers are shown). ^1^ Two clusters of cpb2 (C-1 and C-2; Figure 1) are shown. ^2^ plc gene containing inserion of 834-bp sequence as shown in Appendix A.

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
