# Peer review of "Prevalence and Genetic Diversity of Toxin Genes in Clinical Isolates of Clostridium perfringens: Coexistence of Alpha-Toxin Variant and Binary Enterotoxin Genes (bec/cpile)"

_toxins, 2019, doi:10.3390/toxins11060326_

Round 1

Reviewer 1 Report

In this manuscript, the authors present the results from a survey of a substantial number (n=798) of Clostridium perfringens isolates from clinical specimens.  Information provided suggest that isolates were not epidemiologically linked therefore are less likely to be subject to sampling biases, an observation that is largely supported by the subset presented in Table 3. 

In addition to carrying out toxin genotyping, the study provides some phylogenetic context, and provides a more focused comparison of toxin (nucleotide and/or amino acid) sequences. 

Some aspects require authors attention:

1. The footnote to Table 2 should explain the “+/-“ symbol.  I am assuming this means that strains of toxinotype C, D or E may be either positive or negative for cpe.  As there are zero of these, it is not a major issue however including a short descriptor would mean readers are immediately informed.

2. Table 4 may be better placed in supplement material?

3. The main point concerns the phylogenetic analyses of the cpe and cpb2 toxin sequences (Figure 1).  Most of the branching is not supported by bootstrap >75 - presumably the identity or near-identity of sequences fails to reproducibly place sequences adjacently? 

Related to this, some of the instances (e.g. CP859, CP2) appear to present with some divergences.  It would be important for the authors to confirm in the text that sequences used for phylogenetic analyses for each target gene were of identical length and complete; truncated sequences may present spurious branches.  Depending on outcome of a re-antoxin sequences (alysis, this figure may not be required in main text.

It may be more relevant to remove redundant sequences and phylogenetically assess single examples of each sequence/allele?  This could be an aspect that affects the authors’ interpretation. 

Author Response

Reviewer #1

1. The footnote to Table 2 should explain the “+/-“ symbol.  I am assuming this means that strains of toxinotype C, D or E may be either positive or negative for cpe.  As there are zero of these, it is not a major issue however including a short descriptor would mean readers are immediately informed.

Answer: As pointed out, the “+/-“ symbol represents either positive or negative for cpe. According to the suggestion, this explanation was added as a footnote of Table 2.

2. Table 4 may be better placed in supplement material?

Answer: According to the suggestion, original Table 4 was changed as supplementary material, Table S1 in the revised manuscript.

3. The main point concerns the phylogenetic analyses of the cpe and cpb2 toxin sequences (Figure 1).  Most of the branching is not supported by bootstrap >75 - presumably the identity or near-identity of sequences fails to reproducibly place sequences adjacently? 

Related to this, some of the instances (e.g. CP859, CP2) appear to present with some divergences.  It would be important for the authors to confirm in the text that sequences used for phylogenetic analyses for each target gene were of identical length and complete; truncated sequences may present spurious branches.  Depending on outcome of a re-antoxin sequences (alysis, this figure may not be required in main text. It may be more relevant to remove redundant sequences and phylogenetically assess single examples of each sequence/allele?  This could be an aspect that affects the authors’ interpretation. 

Answer: Thank you for the comments on phylogenetic trees. I agree to the opinion that bootstrap value of >75% may not reproduce branching. Therefore, we constructed phylogenetic tree again by the use of MEGA.7 program (MEGA.6 was used in the original version), and added a sequence of beta toxin as outgroup. As a result, revised tree of cpb2 had the same branching as that of original version, and could get 80% or more of bootstrap values for two clusters. In the revised manuscript, old trees were deleted and the revised cpb2 tree was inserted as Fig.1, and cluster 1 and 2 were assigned to these two clusters. Footnotes were also revised. Four strains including cpb2 prototype in CWC245 and CP412 were not clustered with high bootstrap value (value was 78). Thus, cluster number (name) was not indicated to these strains. Including above findings, paragraph of cpb2 in Result section (2.3.3.) was thoroughly revised.

Sequence data used for cpe and cpb2 were all complete sequence of ORF and had same length. However, for cpb2, sequences of 16 reference strains were partial sequence, because only partial sequences were available in GenBank database. This was described in Materials and Methods section, 4.2, in the revised version.

Original Fig.1a, phylogenetic tree of cpe was deleted from main text, and moved to supplementary material (Fig. S4). Before analysis, we collected cpe sequences from our isolates and GenBank. As a result of phylogenetic analysis, only two lineages were found and sequence identity was extremely high. As reviewer suggested, only a single example would be presented. After consideration, we decided to describe the detection of two lineages in text only, and its evidence, i.e., phylogenetic tree and alignment in supplementary figures.  

Reviewer 2 Report

The manuscript “Prevalence and genetic diversity of toxin genes in 3 clinical isolates of Clostridium perfringens: 4 coexistence of alpha-toxin variant and binary 5 enterotoxin genes (bec/cpile)” describes a comprehensive investigation of the prevalence of different individual toxins from C. perfringens clinical isolates using different genetic tools. The data is well poised, developed and described. There are some minor comments that arose during my review of this submission and that I think could be useful for the audience. The last paragraph of the introduction should stress more the contribution of the work. Methods deserve a deeper explanation. For instance, toxinotype classification should be more extensively explained. Authors refer that they followed the scheme proposed by Rood et al and they created provisional new types. Which are the bases of this classification? The same for the scheme and profile definition by Xiao et al.

Author Response

Reviewer #2

The manuscript “Prevalence and genetic diversity of toxin genes in 3 clinical isolates of Clostridium perfringens: 4 coexistence of alpha-toxin variant and binary 5 enterotoxin genes (bec/cpile)” describes a comprehensive investigation of the prevalence of different individual toxins from C. perfringens clinical isolates using different genetic tools. The data is well poised, developed and described. There are some minor comments that arose during my review of this submission and that I think could be useful for the audience. The last paragraph of the introduction should stress more the contribution of the work. Methods deserve a deeper explanation. For instance, toxinotype classification should be more extensively explained. Authors refer that they followed the scheme proposed by Rood et al and they created provisional new types. Which are the bases of this classification? The same for the scheme and profile definition by Xiao et al.

Answer: Thank you for constructive suggestions. According to the advices, in the revised manuscript, some points of contribution of this study were added in the final para. of Introduction section. In the Materials and Methods section, a section “4.3 Classification of toxinotype and alpha-toxin type” was added, and definitions of toxinotype and alpha-toxin type were described. Toxinotype was determined by conventional scheme (type A-E, ref.46, McDonel’s paper) and also modified scheme by Rood et al. (type F and G). Alpha-toxin types I, IV, V, VIII were classified by definitions written by Sheedy et al. (ref.15) and Abildgaard et al. (ref.16). New subtypes of IV and V in the present study were created, and arbitrarily assigned, based on amino acids at six positions. These classification scheme and definitions were briefly described in the revised manuscript. Xiao’s scheme was applied to MLST genotyping. In the revised manuscript, Xiao’s MLST scheme was briefly explained.  

Round 2

Reviewer 1 Report

The revisions are acceptable.